# A Systematic Review of Design Workshops for Health Information Technologies

**Mustafa Ozkaynak** [1,*] **, Christina M. Sircar** [1] **, Olivia Frye** [2] **and Rupa S. Valdez** [2,3]

1. College of Nursing, University of Colorado, Aurora, CO 80045, USA; christina.sircar@cuanschutz.edu
2. Department of Public Health Sciences, University of Virginia, Charlottesville, VA 22903, USA; opf5bt@virginia.edu (O.F.); rsv9d@virginia.edu (R.S.V.)
3. Department of Engineering Systems and Environment, University of Virginia, Charlottesville, VA 22904, USA
* Correspondence: mustafa.ozkaynak@cuanschutz.edu

**Abstract:** Background: Design workshops offer effective methods in eliciting end-user participation from design inception to completion. Workshops unite stakeholders in the utilization of participatory methods, coalescing in the best possible creative solutions. Objective: This systematic review aimed to identify design approaches whilst providing guidance to health information technology designers/researchers in devising and organizing workshops. Methods: A systematic literature search was conducted in five medical/library databases identifying 568 articles. The initial duplication removal resulted in 562 articles. A criteria-based screening of the title field, abstracts, and pre-full-texts reviews resulted in 72 records for full-text review. The final review resulted in 10 article exclusions. Results: 62 publications were included in the review. These studies focused on consumer facing and clinical health information technologies. The studied technologies involved both clinician and patients and encompassed an array of health conditions. Diverse workshop activities and deliverables were reported. Only seven publications reported workshop evaluation data. Discussion: This systematic review focused on workshops as a design and research activity in the health informatics domain. Our review revealed three themes: (1) There are a variety of ways of conducting design workshops; (2) Workshops are effective design and research approaches; (3) Various levels of workshop details were reported.

**Keywords:** design workshop; participatory design



## 1. Introduction

Participatory approaches are common for designing user-centered health information technologies (HIT) [1–3]. Participatory approaches encourage including the tacit (and often invisible) knowledge of the users [4] by involving a wide variety of users to ensure all user needs are addressed in the design [5]. Design workshops can be an effective way of eliciting end-user participation by actively incorporating and translating valuable input from design inception to completion. A design workshop can be defined as a codesign environment opportunity for a team to cohesively disentangle a specified problem by undergoing a series of group exercises to either initiate or finalize a design, or to ameliorate an obstacle on an existing design. Workshops can be rewarding by bringing relevant stakeholders together to utilize powerful tools and techniques, culminating in the best possible creative solutions.

HIT Workshops can be utilized as a design or research activity. As a design activity, workshops often focus on solving a HIT design problem and support collaboration between designers and users. For example, a workshop can be organized to modify an existing HIT app (e.g., food tracking app) previously tailored to a specific population to be inclusive of additional populations. As a research activity, workshops can generate and delineate the necessary knowledge to improve the efficacy of a design process and outcome. For

example, a workshop can examine the impact of artificial intelligence on clinical decision making by diverse types of clinicians.

Workshops can be particularly beneficial for the design of HIT targeting diverse end-users (clinicians, caregivers, patients), as the workshop atmosphere can cultivate the conception of a design that meets the needs of all users. As health care becomes increasingly distributed among clinical and daily living settings [6–10], the scope of HIT and its user panel have significantly broadened. Workshops can play a vital role in the redesign of current HIT to match the new scope and expanding user needs.

The benefits of a workshop are contingent on involving the correct selection of participants and utilizing the right tools and techniques for the targeted population. Moreover, conceptual frameworks are useful to keep the scope of workshop activities focused. The aim of this systematic review was to identify the tools, techniques, and approaches available. This information should provide guidance to HIT designers and researchers in devising and organizing design workshops. This guidance could help structure the coordination and delivery of a more efficient workshop targeting user-centered design practices. Furthermore, examining the strengths and weaknesses of previous workshop would serve as a lesson to design more novel workshops. For example, at the time of COVID-19 or a similar pandemic, workshops could be held remotely.

## 2. Methods

A systematic literature search was conducted by a health librarian in five databases (Medline, Embase, Cochrane Central Register of Controlled Trials, Cochrane Database of Systematic Reviews, and Web of Science) including the following keywords: Participant design, codesign, cooperative design, user involvement, stakeholder participation, health information technology, medical informatics, software, phone app, mHealth, digital health, games, gaming, gamify, telehealth, telemedicine, software design, universal design, computer-aided design, mobile applications, stakeholder, patients, health provider, physician, nurse, therapist, caregiver, user, and family. (Supplementary Material File S1 describes search strategies).

The search process identified a total of 568 articles, with 562 articles remaining after initial article duplication removal. Figure 1 shows the Preferred Reporting Items for Systematic Reviews and Meta-Analyses (PRISMA) [11] diagram for the search methodology. The diagram outlines the number of records that were identified, included, and excluded.

The second and third authors (C.M.S. and O.F.) conducted concurrent independent reviews of the title field by selecting articles that included participatory design (PD), co-design (CoD), and/or were related to HIT, reducing the number to 364 articles. The concurrent independent reviews continued for the reconciliation of abstracts and pre-full-text reviews (i.e., reviewing full text partially, focusing on only predetermined sections), applying inclusion/exclusion criteria, and resulting in 114 and 72 relevant records, respectively. Studies that met inclusion criteria were journal articles and conference papers explicitly mentioning PD or codesign sessions and workshops utilized in HIT. The first author (M.O.) mediated unreconciled article disputes following each screening cycle. A full-text review of 72 articles was completed by the second author (C.M.S.), followed by the extraction of the following elements from the included articles: year, title, author, journal, type of article, country, purpose, target technology, target disease/medical population or service, guiding framework, phases, participant selection, participant profiles, number of participants, workshop duration, number of workshops, activities and approach, tools and equipment, deliverables/results, evaluation, pre and post PD methods, issues reported/weaknesses, and strengths/benefits.

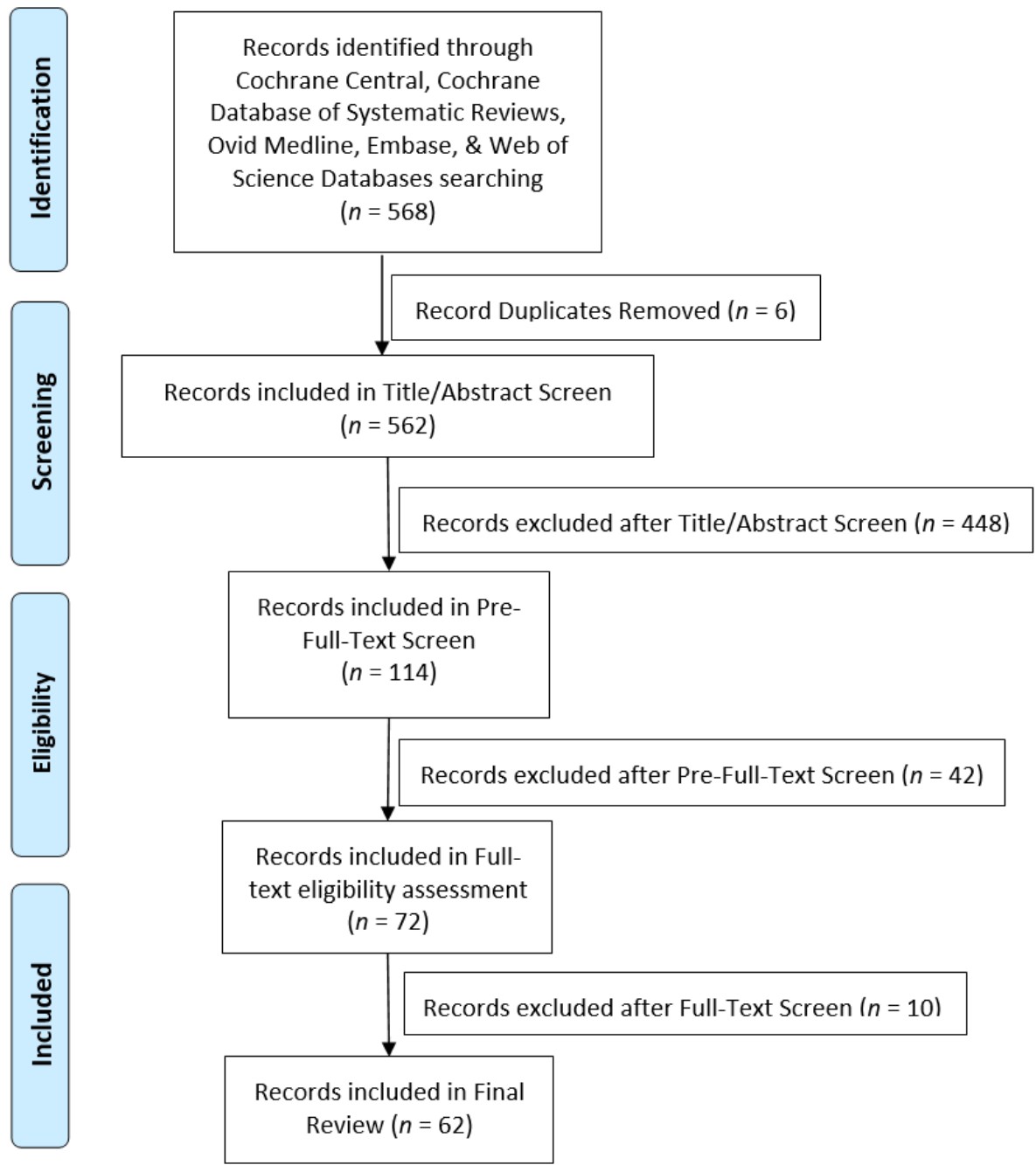

**Figure 1.** PRISMA diagram for search methodology.

Studies that were excluded were those that mentioned PD sessions/workshops but did not explicitly describe the implementation process, those that were not focused on a health-related field/topic, and/or those that did not differentiate between PD methods. Abstracts without a full paper, posters, and duplicates were likewise excluded. The reconciliation of abstract reviews resulted in 250 excluded records, leaving 114 articles for pre-full-text review. The pre-full-text review resulted in excluding 42 articles, and an additional 10 articles were excluded in the full-text review. Sixty-two articles published from 2006 to 2020 met the inclusion criteria.

Publications that were included met the following criteria: (1) written in English, (2) author identified, (3) targeted medical/health-related technology, (4) followed a participatory approach to workshop or session design, (5) full-text available electronically, and (6) included participatory design interchangeable terms such as cocreation, codesign,

and user-centered design. Eligible publications included records with varied stakeholders (e.g., clinicians, patients, caregivers), and involved adults and/or children participating in HIT PD. Final full-text inclusion articles ranged in publication date from 2006 to 2020. Additionally, studies that summarized a participatory process without clearly detailing the activities and/or actions were excluded.

## 3. Results

Sixty-two publications [12–73] from the years 2006 to 2020 were included in the review. These publications are a result of 59 unique studies. Included publications were predominantly journals (*n* = 57), but conference papers were also included [36,55,60,63,71] (*n* = 5). Studies were conducted in six continents (Europe, 34; Oceania, ten; North America, eight; Asia, four; Africa, one; South America, one). Study location was not reported in four studies. Included studies focused on consumer-facing health technologies (e.g., app or website; *n* = 39), clinical health information technologies (e.g., electronic health records; *n* = 8), technologies that involve both clinicians and patients as users (e.g., telehealth technologies; *n* = 12) and other technologies (*n* = 3). The studies represented a wide variety of health conditions (e.g., mental health, heart failure, asthma, dementia, kidney transplantation), life span (e.g., youth, adult, elderly), and type of users (patients, caregivers, clinicians). Workshops could be accomplished as a single workshop (*n* = 10) or as a series (*n* = 52). The highest number of reported workshops in a single study was 20 [26]. Workshops could include homogeneous (e.g., only clinicians) or heterogenous (e.g., clinicians and patients) groups. The number of participants (in each workshop) varied from 4 [17,19,23] to 47 [52]. The duration of a workshop varied between one hour and over one day.

The included studies focused on a wide variety of target diseases, settings, treatments, or populations. In three of the studies, the target population consisted of clinicians and other stakeholders as users. In four of the studies, the target population focused on the caregivers of patients with various conditions. The remaining 55 studies focused on various conditions. Antibiotic management was the focus in three studies. Each of the following conditions was the focus in two included papers: Type I Diabetes, asthma, adolescent mental health, HIV, chronic illness (general), rheumatoid arthritis, heart disease, and mental health (general).

Many workshop activities and deliverables were reported. These activities include:

### 3.1. Discussion Activities

- Whole group or small group discussions (guided by semistructured questions or findings from previous steps)
- Brainstorming (using Post-it notes, poster size papers, flipcharts)
- Affinity diagrams
- Collecting ideas
- Sorting methods
- Idea notetaking on sticky Prioritization
- Exploring a selected technology
- Creating human scatter graph
- Note cards

### 3.2. Description of Experience

- Personas
- Vignettes on a 'story-board' in cartoon-strip format
- Case vignettes
- Scenarios
- Journey mapping
- Storytelling/storyboards
- Arranging pictures and labels describing the stages of before, during, and after a visit
- Talking on a specific experience

- Creating collages
- Creating instant visuals

### 3.3. Prototyping

- Developing mock-up models or prototypes
- Frame-by-frame sketches
- Solution sketch
- Presentation of mock-up, etc., by participants
- Rapid cycle iterative design
- Design sprints

### 3.4. Creating Conceptual Representations

- Concept mapping
- Responding to questions by creating a human scatter graph
- Creating an instant visual of the group's perception and experience of mobile games and games for health
- Sketches of the participants' design concepts
- Participant narrative representation of thoughts

### 3.5. Evaluation Activities

- Technology/product demonstrations
- Hands-on use of technology
- Debriefing
- Technology/prototype/idea/solution evaluation and providing feedback
- Workshop feedback
- Applications/apps
- Think Aloud
- Walkthrough exercise
- Questionnaires

### 3.6. Presentations

- Presentation of results of previous phases/literature, etc., by moderators
- Presentations or giving talk on a specific topic

### 3.7. Game Playing

- Design games
- Role-playing

### 3.8. Stimulate Group Participation

- Design cards
- Icebreaker session
- Field kit
- Prompt cards
- Presented substance images
- Lightning Demos activity

The grouping of workshop activities is not necessary mutually exclusive. Moreover, some of these activities may overlap. Primary deliverables of these workshops were:

- Prototypes/mock-up models (paper-based or wireframes)
- Research themes
- A list of recommendations/solutions/ideas
- Product/technology evaluation

Table 1 highlights the abstracted information from each study including authors, publication type, participants, activities, and deliverables.

**Table 1.** Summary of Results of 62 publications.

| Authors | Target Disease, Setting, Treatment or Population | Participants | Number of Workshops | Activities | Deliverables |
|---|---|---|---|---|---|
| Amann et al. 2020 [12] | Spinal cord injury | 2 Researchers, 2 Designers, 4 Clinicians, 5 Adult Patients | 1 | Personas, design sprints | A prototype in the form of a clickable user interface |
| Aufegger et al. 2020 [13] | Young patients who are about to undergo a complex intervention | 14 Children 11 Adults | 1 | Journey mapping, redesign journey mapping | A journey map visual |
| Burford et al. 2015 [14] | Diabetes Mellitus (DM) Type II | 4 Researchers, 10 Clinicians, 1 Practice manager | 2 | Ideas collected and affinity diagram | A list of recommendations for patients' empowered behavior and how tablet devices can support it |
| Castensøe-Seidenfaden et al. 2017 [15] | DM Type I | 33 Young patients, 18 Parents, 18 Clinicians, 45 experts | 7 | Ideas collected, prioritized, sketching prototypes, Post-it notes, flip charts | Research themes |
| Castro-Sánchez et al. 2019 [16] | Antibiotic management | 29 experts | 1 | Small and whole group discussion of previously determined questions | A list of implementation, adoption and evaluation related threats and solutions |
| Wan Sze Cheng et al. 2018 [17] | Men's Mental Health and Well-Being | 40 Researchers and students | 6 | Whole group discussion, presentations, and sketching | Prototype |
| Curtis & Brooks, 2020 [18] | Nursing Homes | 10 Clinicians | 2 | Presentation from previous steps of design and whole group discussion | A list of recommendations for (i) implementation; (ii) Sustaining engagement; (iii) Transforming care |
| Danbjørg et al. 2018 [19] | Osteoarthritis | 4 Adults | 1 | The starting point was the cultural probes (from the test phase), which served as the opening to hear about the participants' experiences; subsequently, the participants brainstormed and created a mock-up model | Evaluated paper-based mock-ups |
| Davis et al. 2018 [20] | Asthma | 13 Young patients | 1 | Talking on app experience, creating collages, creating concept maps | Prototype app |

**Table 1.** *Cont.*

| Authors | Target Disease, Setting, Treatment or Population | Participants | Number of Workshops | Activities | Deliverables |
|---|---|---|---|---|---|
| Fleming et al. 2019 [21] | Adolescent Mental Health | 31 Young patients | 2 | Discussion, mock-up on paper, storytelling and brainstorming on sticky notes, posted them on a wall | Research themes |
| Garne Holm et al. 2017 [22] | Neonatal homecare | 6 clinicians 4 researchers 1 IT consultant 10 parents 5 infants | 2 | In the first workshop, cases developed from the first phase of the study were presented; the participants provided solutions to each of the cases presented. In the second workshop, technology was tested in a simulated environment | Research themes |
| Giordanengo et al. 2018 [23] | DM Type I | 4 clinicians 5 patients | 2 | Whole group discussions | Research themes |
| Giroux et al. 2019 [24] | Care givers of seniors | 18 Clinicians 30 Caregivers 26 Community workers | 8 | Sorting method; brainstorming; persona; paper prototyping; group discussions | Prototype website, research themes |
| Gonsalves et al. 2019 [25] | Adolescent Mental Health | 46 Adolescents | 3 | Exploring a selection of popular games and apps; story building to create personas and problem scenarios; paper prototyping; discussion about prototype ideas; brainstorming; discussion | Evaluated prototype |
| Gordon et al. 2016 [26] | High-risk women after discharge from hospital | 4 patients 4 clinicians 2 social workers 1 clinic administrator 3 support staff 2 research staff 1 programmer | 20 | Rapid cycle iterative design | Three applications |
| Greenhalgh et al. 2015 [27] | Assisted living settings | 40 residents 14 service providers 7 technology suppliers | 10 | Vignettes on a 'story-board' in cartoon-strip format | Research themes |
| Grenha Teixeira et al. 2019 [28] | EHR Stakeholders | 67 stakeholders | 2 | Concept Map creation for a service | Concept Map |

**Table 1.** *Cont.*

| Authors | Target Disease, Setting, Treatment or Population | Participants | Number of Workshops | Activities | Deliverables |
|---|---|---|---|---|---|
| Hemingway et al. 2019 [29] | HIV | 18 individuals | 2 | Creating a human scatter graph, creating instant visuals, whole group, and small group discussions and brainstorming | The list of recommended game features |
| Hobson et al. 2018 [30] | Motor neurone disease | 3 patients 6 caregivers 1 clinician | 2 | An icebreaker session; patient journey mapping; arranging pictures and labels describing the stages of before, during, and after a visit; personas; whole group discussion | Journey map |
| How et al. 2017 [31] | TBI | 8 clinicians | 2 | Design cards and field kit | Conceptual ideas for TBI cognitive telerehabilitation |
| Jeffery et al. 2017 [32] | Nurses | 20 nurse participants | 3 | Video vignette, creating sketches, debriefing | Debriefed sketches |
| Jensen et al. 2018 [33] | Hip fractures | 42 various participants | 5 | Present findings from previous study, brainstorming using Post-its and board, prototyping | Prototype |
| Jessen et al. 2018 [34] | Chronic illness | 22 participants aged 17-64 | 6 | Design games, scenario making, prototyping, and brainstorming. | Research themes |
| Jessen et al. 2020 [35] | Chronic illness | 22 participants aged 17-64 | 6 | Design games, prototyping, and scenario making | Prototype; suggestions |
| Klemets & Toussaint, 2015 [36] | Nurses | 9 nurses | 2 | Scenario making, role-playing | Artifacts in the form of scenarios and a prototype |
| Kocaballi et al. 2020 [37] | Primary care consultations | 16 general practitioners 2 researchers | 3 | Affinity diagramming, brainstorming, and prototyping | Research themes |
| Latulippe et al. 2020 [38] | Caregivers of functionally dependent seniors | 74 adult participants 4 researchers | 11 | Whole group discussions | Research themes |
| Latulippe et al. 2020 [39] | Caregivers of functionally dependent seniors | 74 adult participants 4 researchers | 11 | Presentations, sorting, whole group discussions, brainstorming, prototyping, sketching, and pretesting | Research themes |

**Table 1.** *Cont.*

| Authors | Target Disease, Setting, Treatment or Population | Participants | Number of Workshops | Activities | Deliverables |
|---|---|---|---|---|---|
| Lee et al. 2018 [40] | Rheumatoid arthritis | 10 adult patients<br>18 health care professionals | 3 | On hands use of a technology and debrief | Not reported |
| Lundin & Mäkitalo, 2017 [41] | Hypertension | 15 adult patients<br>1 project leader/moderator<br>1 company representative<br>1 researcher<br>1 video staff | 3 | Whole group discussion, product demonstration and hands-on practice, debrief | Research themes |
| Lupton, 2017 [42] | Generic | 25 adult participants | 1 | Concept mapping, brainstorming, storyboard creating | Research themes |
| Marent et al. 2018 [43] | HIV | 61 clinicians<br>77 adult participants | 14 | Discussion | Research themes |
| Martin et al. 2020 [44] | Adolescents | 74 adolescent participants | 1 | Product demonstration and hands on activities | Product evaluation |
| Martin-Hammond et al. 2019 [45] | Seniors | 18 adult participants<br>2 researchers | 1 | Whole group discussion, small group discussion, scenario, sketches of the participants' design concepts, brain storming, affinity diagrams | Research themes |
| Moen & Smørdal, 2012 [46] | Rare conditions | 50 participants | 15 | Discussion | Research themes |
| Naeemabadi et al. 2020 [47] | Total knee replacement | 8 participants<br>2 physiotherapists<br>1 nurse<br>1 orthopedic surgeon<br>3 researchers<br>4 student assistants<br>2 software developers | 2 | Brainstorming, discussions, paper prototyping, prototype evaluation | Preliminary paper prototypes, research themes |

**Table 1.** *Cont.*

| Authors | Target Disease, Setting, Treatment or Population | Participants | Number of Workshops | Activities | Deliverables |
|---|---|---|---|---|---|
| Nielsen et al. 2020 [48] | Kidney transplantation | 9 clinicians<br>4 patients<br>2 family members<br>3 researchers<br>2 members of kidney association<br>1 dietician<br>1 physiotherapist<br>6 others | 2 | Brainstorming | A prototype app. |
| Noergaard et al. 2017 [49] | Heart disease | 7 patients<br>3 clinicians<br>2 systems architects<br>3 moderators<br>3 observers | 3 | Questionnaires, hands-on exercises, group discussions, plenary discussion, presentations | Work-in progress reports |
| Ospina-Pinillos et al. 2019 [50] | Mental health | 10 young adults<br>7 health professionals | 2 | Mock-ups and end-user sketching | Workshop discussion notes and 208 artifacts |
| Ospina-Pinillos et al. 2020 [51] | Mental health | 7 young adults<br>11 health professionals | 2 | Discussion, review of mock-ups, hand-draw ideas | Handwritten notes, 194 source documents were developed and analyzed (2 sets of workshop notes and 192 artifacts produced by participants) |
| Peiffer-Smadja et al. 2020 [52] | Antibiotic management | 47 health professionals | 1 | clinical scenarios, discussion | Electronic questionnaire, research themes |
| Peiris-John et al. 2020 [53] | Adolescents | 8 adolescents<br>3 young adults<br>5 digital health care providers<br>6 community stakeholders<br>9 researchers | 2 | small group discussion, brainstorming | Data/feedback used to create prototype |
| Peters et al. 2017 [54] | Asthma | 13 young patients | 4 | collaborative collage, individual concept mapping, and paper prototyping | Collages, concept maps, and paper prototypes |

Table 1. *Cont.*

| Authors | Target Disease, Setting, Treatment or Population | Participants | Number of Workshops | Activities | Deliverables |
|---|---|---|---|---|---|
| Pollack et al. 2016 [55] | Generic | 3 researchers<br>11 physicians | 1 | Presentation, group brainstorming, discussion, mock-ups, presentation of mock-ups | Notecards, whiteboard recording of discussions, crafted handmade designs |
| Jakobsen et al. 2018 [56] | Osteoporosis | 2 researchers<br>6 female patients<br>5 clinicians<br>5 experts | 3 | Games, brainstorming, mock-up, wireframe review, | Mock-ups, field notes, pictures, video recordings |
| Rawson et al. 2018 [57] | Antibiotic management | 30 adult patients | 2 | Discussions | Research themes |
| Revenäs et al. 2014 [58] | Rheumatoid arthritis | 5 adult patients<br>3 experts<br>2 researchers | 4 | Discussion of previous focus groups, warm-up session, brainstorming | Data collection from an online notice board, interactive boards, Post-it notes on plastic sheets, video recordings, observation protocols |
| Robinson et al. 2009 [59] | Dimentia | 24 patients<br>13 carers | 5 | Presented with list of priorities from scoping stage for discussion, scenarios, presented a range of existing devices for discussion | Prototypes, research themes |
| Ruland et al. 2006 [60] | Children | 12 children aged 9–11yo | 8 | Role play and scenarios, prototyping, program/game testing using think aloud | Prototypes, observation notes, videotape |
| Scandurra & Sjölinder, 2013 [61] | Seniors | 8 adults aged 65–80 | 7 | Questionnaires, iterative discussion from previous workshop data | Researcher notes, questionnaires |
| Sin et al. 2019 [62] | Psychosis | 3 patients<br>3 family members<br>1 clinician<br>1 voluntary service lead.<br>6 researchers | 4 | Review of previous studies, brainstorming, sketching, draft wireframing, walkthrough exercise | Draft hand-sketched plans and wireframes, mock-ups of Web pages, and source materials for the intervention, intervention prototype, themes |
| Swallow et al. 2016 [63] | Seniors | 33 participants aged 55–85 | 4 | Questionnaire, presentation from previous study, affinity diagrams, presentation of results, discussion | Audio recording, affinity diagram, notes, Post-it notes, questionnaires |

**Table 1.** *Cont.*

| Authors | Target Disease, Setting, Treatment or Population | Participants | Number of Workshops | Activities | Deliverables |
|---|---|---|---|---|---|
| Terp et al. 2016 [64] | Schizophrenia | 4 young adults<br>7 healthcare providers<br>6 experts | 10 | Storyboard, card sorting, mock-ups, paper prototypes | Hand-drawn workshop invitation, workshop preparation descriptions, workshop notes, written reflections, group interviews |
| Tremblay et al. 2019 [65] | Caregivers of Functionally Impaired Seniors | 4 researchers<br>11 caregivers<br>16 community workers<br>11 health and social service professionals | 4 | Group discussion, brainstorming, paper prototypes | Audio and video recordings, artefacts, paper documents, spreadsheets |
| van Besouw et al. 2015 [66] | Aural rehabilitation | 28 adult participants<br>2 researchers<br>3 experts | 9 | Presentation, discussion, mock-ups | Feedback and observations incorporated into a prototype music rehabilitation program |
| Wannheden & Revenäs, 2020 [67] | Parkinson's disease | 7 patients<br>4 neurologists<br>3 nurses<br>2 physiotherapists | 4 | Note cards, group discussion | Data from notecards and focus group discussions, research themes |
| Warren et al. 2019 [68] | EHR stakeholders | 48 participants | 2 | Presentation, mock-ups, small group vignette activity, group presentations and feedback, debrief | Quantitative and qualitative questionnaires |
| Wherton et al. 2015 [69] | People with assisted living needs | 61 participants | 10 | Case vignettes, case narratives, discussion, flow-diagram, presentation, prompt cards, storyboards, narratives | Research themes |
| Winterling et al. 2016 [70] | Cancer patients with sexual problems and fertility distress | 10 former patients<br>2 significant others | undetermined | Discussion, ice breaking, mock-up creation, discussion, prototype | Mock-up, prototype |

**Table 1.** *Cont.*

| Authors | Target Disease, Setting, Treatment or Population | Participants | Number of Workshops | Activities | Deliverables |
|---|---|---|---|---|---|
| Woods et al. 2018 [71] | Heart disease | 6 clinicians<br>1 patient | 2 | Prototypes presented for feedback and improvement cycles, Lightning Demos activity, brainstorming, personas, solution sketch, comic-like storyboard, brainstorming | 14 frames of sketches, labels and descriptions, posters, wireframes, initial software build |
| Xu et al. 2020 [72] | Caregivers of children with atopic dermatitis | 20 caregivers<br>10 healthcare providers<br>4 digital health experts | 3 | Discussion, sketching | Sociodemographic questionnaire, technology acceptance questionnaire, workshop evaluation form, field notes, observation logs, photos, written products, audio recordings; research themes |
| Zhang et al. 2019 [73] | Substance use disorders | 10 patients<br>10 health care professionals | 3 | Participant narrative representation of thoughts, brainstorming, presentation on gamification approaches, idea notetaking on sticky notes, whole group discussion, Sketching, presentation of substance images. | Audio recording, prototype sketches, common element identification from data |

Some workshops were performed as stand-alone design and research activities, while others included workshops as one part of a multiphase design and research process, encompassing other data collection methods such as interviews, focus groups, or observations. Workshops included in a multiphase project utilized preceding activity outcomes to preface the current workshop. Likewise, following an iterative process, workshops provided input to guide subsequent design activities, unless the design workshop was the concluding activity.

Some workshops were guided by conceptual or methodological frameworks. Reported conceptual frameworks included behavior change theories [12,29], perspectivist theory [14], democratic dialog theory [14], user experience framework [24], Ottawa decision support framework [26], medical research council framework [30], theoretical framework of social justice [38,39], self-determination theory [44,47,54], IDAS framework [44], nudging theory [44], and cultural historical activity theory [46]. Methodological frameworks included cooperative inquiry [13], appreciative inquiry [18], hermeneutics philosophy [19], ethnography [22], double diamond design process [35], action research [33,48,56,58], PICTIVE [55], and cooperative design [61]. Some workshop studies included in this review were guided by custom frameworks [32,40,59]. The dominant data analysis method was qualitative thematic analysis. Other inductive qualitative methods were also reported. Seven articles reported the use of multiple or mixed methods [15,52,57,67,68,72,73]. The majority of the workshops were audio and video recorded.

Although uncommon across studies, some distinct phases within or between workshops were identified. One study [22] organized two workshops and distinguished them as "creative" and "technical" workshops. Another [28] identified the following phases: exploration, ideation, reflection, and implementation. Jeffery et al. [32] identified three phases: priming, designing, and debriefing. Yet another [35] included the following phases: discovering, defining, developing, and delivering. One [59] identified three phases: scoping, participatory design, and prototyping. One study [61] identified four phases: user needs assessment, low fidelity prototyping, high fidelity prototyping, and functional prototyping. One [17] study identified three phases: Discovery; Evaluation; Prototype. Multiple studies used the phases from inception to implementation consistent with a system development life cycle: identification of needs, development/prototyping, and evaluation.

The level of workflow details reported in the included studies varied. Of the 62 studies, 11% (*n* = 7) described a formal evaluation, 60% (*n* = 32) a guiding framework, 63% (*n* = 39) the duration of the workshop, and 85% (*n* = 53) the participant selection as being critical components when planning a workshop but were not consistently reported.

Only seven [33,40,61,63,66,68,72] publications included data on the evaluation of the workshop. However, only five of them [61,63,66,68,72] reported a deliberate effort for evaluation. These efforts include asking verbal feedback within the whole group and/or a questionnaire. The other two collected feedback informally or extracted workshop evaluations indirectly from qualitative data that was originally collected for the design of focus. In the provided feedback, the evaluation of workflow could be intertwined with the evaluation of the design workshop focus. Table 2 shows a summary of the workshop evaluations.

**Table 2.** Workshop evaluation results.

| Authors | Evaluation Summary |
|---|---|
| Jessen et al. 2018 [34] | "Overall, we can conclude from the vast variety of user inputs that the workshops were successful in generating new and creative concepts and ideas for mHealth tools … The participants and the facilitators alike found the workshops to be both productive and enjoyable. In fact, when getting feedback at the end of the first workshop, all 3 groups wanted to spend more time on the next workshop". |
| Lee et al. 2018 [40] | "Patients, health care professionals, and managers confirmed the relevance and value of the overall concept as well as the organizational setup". |
| Scandurra & Sjölinder, 2013 [61] | "At the concluding workshop the participants described their overall experiences, both with respect to the latest version of the device and with respect to the overall impressions about the project". |
| Swallow et al. 2016 [63] | "The workshop concluded with a general discussion in which the research team summarized the overall findings and gave participants the opportunity to provide any additional feedback about the problems and solutions, as well as their experience of taking part in the participatory design workshop". |
| van Besouw et al. 2015 [66] | "The method used to collect feedback during the trial (an online survey completed by users at the end of each session) also resulted in 'honest' feedback that captured the users' immediate and unrestrained reactions to the session and software". |
| Warren et al. 2019 [68] | "Quantitative evaluation questionnaires were undertaken by 43 participants. Non-responses to individual questions were excluded from analysis. Responses from the evaluation questionnaire indicated that participants found the workshop process used for this project to be enjoyable, useful and interesting. Participants indicated that this workshop stimulated their interest in being involved in future healthcare design work, further emphasizing the potential value of the methods described in healthcare research and development". |
| Xu et al. 2020 [72] | "The workshop evaluation comments indicated that the co-design workshop was successful in creating and generating new ideas and content for smartphone app development". |

In summary, this study resulted in three important findings. First, the list of activities in conjunction with Table 1 highlighted that designers and researchers have various approaches to conduct workshops. Second, although the included studies reported rich findings overall, a lack of consistent evaluations prevented the ability to compare the effectiveness and appropriateness of workflow activities for different purposes. Third, workshops could possibly be utilized more frequently and fastidiously if the many workshop details are disseminated. These findings led to three themes as described in the discussion.

## 4. Discussion

This systematic review focused on workshops as a design and research activity in the domain of health informatics. This type of participation (i.e., workshops) presupposes the need to give voice to the users, rather than users just serving as a source of information or observation. Our review revealed three themes: (1) There are a variety of ways of conducting design workshops; (2) Workshops are effective design and research approaches; (3) Various levels of workflow details were reported. These themes can provide important insights on HIT development by providing a guidance to researchers and designers to operate and disseminate design workshops in a systematic and effective way.

### 4.1. There Are a Variety of Ways of Conducting Design Workshops

Designers and researchers have numerous workshop activity options as listed in the results section. Sanders et al. [74] identified three classes of workshop activities: (1) making tangible things; (2) acting, enacting, and playing; (3) talking, telling, and explaining. The workshops on the design of health informatics interventions have utilized approaches from all three classes. The selected studies do not provide sufficient details to assess whether any workshop activity is better than another, or if a specific workshop activity is superior (or inferior) in exploring a target technology design, participant characteristics, or other factors. However, we argue that the selected approach should be congruent with (1) the design problem, (2) participant characteristics, and (3) available resources. We also argue that a combination of multisensory eliciting activities (visual, auditory, tactile) within the same workshop will more thoroughly engage the participants, allowing for better absorption of material and understanding. This review is useful in terms of balancing tradeoffs between diverse workshop activities and combining multiple complimentary activities to further strengthen participant acumen and workshop efficaciousness.

Various participants may be more productive using different activities. For example, select participants can be more productive in making tangible things (e.g., prototyping), some participants can be more effective in acting and playing (e.g., role playing); and likewise, other participants can offer more dynamic input in talking and discussion (e.g., brainstorming) activities. Therefore, employing multiple types of workshop activities can best utilize diverse participant characteristics, therefore yielding an overall more productive workshop. Moreover, various participant characteristics such as disability status, developmental level, language barriers, and cultural sensitivities should be accounted for when selecting workshop activities.

Various conceptual and methodological frameworks have been used in organizing workshops. However, the contributions of the highlighted frameworks to the design process were implicit. Although there are frameworks specific to workshops (e.g., [75]) the adoption of these frameworks by workshop designers and researchers is ambiguous. Future studies should focus conceptual and methodological frameworks that link the research questions and design objectives to the workshop activities and deliverables.

### 4.2. Workshops Are Effective Design and Research Approaches

Participatory design suggests involving users throughout the entire design process. Design workshops should be a part of the broader participatory design process, in that they are emblematic of the values of this overall philosophy, creating a space in which designers and users can work together collaboratively to formulate design solutions. The success of a design workshop can be measured by its ability to bring out invisible or tacit knowledge. This study provides designers and researchers a variety of options to facilitate the capture of this hidden knowledge. As the workshops are more systematically evaluated, which activity is more suitable for different purpose and context will be better understood.

None of the included studies cautioned against identified weaknesses or lack of effectiveness and efficiencies of the design workshop. However, we present gaps both in terms of how such workshops have been evaluated and assessing the effectiveness of different types of strategies for different contexts.

Design workshops can improve the design of interventions that affect various sub-domains of health informatics. However, these workshops focus on varied aspects of design, not necessarily examining implementation issues. Implementation workshops can complement design workshops and support adoption and sustainability of the informatics interventions that were developed within design workshops.

### 4.3. Various Levels of Workshop Details Were Reported

Replicating successful design workshop practices depend on disseminated details. The papers reported many details but lacked consistency across studies. Moreover, there are some details that were not reported by any studies. For example, all 62 studies included the

overall number of participants in a workshop series but did not consistently report on the exact number of participants for each workshop, expertise, and demographic information of participants. Moreover, none of the studies provided full details on the selection of sample; how diversity, equity, and inclusion (DEI) issues (e.g., inclusion of vulnerable groups) were addressed; and any potential sampling bias.

We argue that any study that employs and reports on design workshops should report: the purpose, number of participants, selection of sample, how DEI issues are addressed, workshop duration, workshop agenda, details of conducted activities (e.g., activity identification, selection criteria), guiding framework(s) specific to the workshop (if applicable), expected and actual deliverables, and workshop evaluation process and results.

### 4.4. Recommendations for Conducting a Workshop

Based on our review, we developed the following recommendations for researchers and designers for organizing design workshops:

- The preparation/planning stage is critical for the success of the project. It is important to be cognizant of differing levels of aptitude. In some cases, participants may benefit from preworkshop technology education (e.g., brochures/pamphlets, emails) to elicit a better understanding of the technology or concepts novel to the participant, and subsequently support workshop participation preparedness.
- Well defined research/design questions/objectives should be the main drivers of other decisions related to organizing the workshop: participants, technology or intervention being designed, conceptual and methodological framework used, workshop activities employed. There should be a congruence among (1) research/design questions/objectives, (2) sampling, (3) selected activities, (4) technology/intervention that is being designed, (5) guiding framework, and (6) available resources.
- Sampling should reflect a wide range of user needs. Vulnerable populations should particularly be considered.
- An introduction may include an ice breaker/warm up exercise to establish commonality between participants and cultivate a trustful atmosphere with the facilitator.
- Workshops could benefit from a facilitator/moderator and a dedicated individual who will document the workshop activities and outcomes by taking notes or audio/video recording. Facilitators should be mindful of potential power imbalances in varied stakeholder groups that can result in a dominating one-sided perspective.
- A synergistic creating process can benefit from a relaxed environment, allowing for participants to freely move about and take breaks as needed. Providing coffee, snacks, or meals and encouraging a flexible atmosphere may encourage willingness for continued participation in an often time-intensive proceeding.
- Utilizing diverse activities will more likely provide better engagement and input, particularly for the heterogenous groups.
- Workshop conclusions should include a formal evaluation (e.g., an exit questionnaire, brief interviews, providing a visible note taking board to post feedback throughout the workshop) to provide structured feedback when the workshop findings are disseminated. If a formal evaluation is not feasible, the workshop may include debriefing or collective reflection to discuss participant experiences attained from design activities. This exercise dually acts as an informal evaluation of the successes and/or areas in need of improvement and validates the importance of participant contributions to the participant themselves.

These recommendations are the result of critiquing the literature and discussion with the research team. While this is not a comprehensive list of approaches, it provides our recommendations from the 62 studies. Included studies reported on in-person workshops. However, our recommendations potentially hold for conducting workshops remotely using computer mediation due to pandemics (e.g., COVID-19) or any situation that makes gathering impossible or impractical. Novel workshop activities specific to or more effective for remote workshops may be needed.

*4.5. Limitations*

A meta-analysis of study results was not possible because of the heterogeneity of the design and study results. We therefore provided a descriptive analysis of publications. We acknowledge that the retrieved studies do not necessarily represent a comprehensive list of all HIT workshops reported in the literature but list studies from the scientific literature returned during our search and that met our inclusion criteria.

**Supplementary Materials:** The following are available online at https://www.mdpi.com/article/10.3390/informatics8020034/s1, File S1: Detailed search strategies.

**Author Contributions:** Conceptualization, M.O., C.M.S. and R.S.V.; review, M.O., C.M.S. and O.F.; formal analysis, M.O. and C.M.S.; writing—original draft preparation, M.O. and C.M.S.; writing—review and editing, M.O., C.M.S., R.S.V. and O.F. All authors have read and agreed to the published version of the manuscript.

**Funding:** This research received no external funding.

**Institutional Review Board Statement:** Not applicable.

**Informed Consent Statement:** Not applicable.

**Data Availability Statement:** Data Sharing not applicable.

**Acknowledgments:** The authors thank Lilian Hoffecker for searching the literature and Suzanne Lareau for editorial support.

**Conflicts of Interest:** The authors declare no conflict of interest.

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
