# Peer review of "A Systematic Review of Design Workshops for Health Information Technologies"

_informatics, doi:10.3390/informatics8020034_

Round 1

Reviewer 1 Report

Thank you for sharing interesting work on Design Workshops. More insights into how such activities can be organized and contribute to ensure good and usable health information technology solutions are important. With the ambition of giving guidance for how to organize such workshop activities, I would like to see a few improvements, suggesions and clarification to the present work.

  • choice of sources; the chosen DBs are largely capturing material from the health section, and since Design Workshops is an established strategy in information systems development, I would like to see search in DBs that capture this material
  • elaboration of how the 3 themes was revealed/synthesized before presenting in the Discussion section
  • group the LONG list of activities - starting on line 147, p.4 ff.
  • discuss the 3 themes -and probably include considerations related to the motivation  of the review stated as "capacity to bring out invisible/tacit", and how considering the 3 themes can give bring in important insights to the health information tech development
  • elaborate how the section on "Recommendations" comes out of the systematic review - right now it reads more like experiences of the authors.

Author Response

Thank you for sharing interesting work on Design Workshops. More insights into how such activities can be organized and contribute to ensure good and usable health information technology solutions are important.

  • We thank the reviewer for the encouraging words.

With the ambition of giving guidance for how to organize such workshop activities, I would like to see a few improvements, suggestions and clarification to the present work.

choice of sources; the chosen DBs are largely capturing material from the health section, and since Design Workshops is an established strategy in information systems development, I would like to see search in DBs that capture this material

  • We appreciate the suggestion; however, this study has a particular focus on design workshops for health information technologies (HIT). Given the number of full papers reviewed (n=62), we felt there was sufficient data in this one area of HIT. Supporting notion, the breadth and depth of the search was sufficient to draw useful conclusions on design workshops. Moreover, “Web of science,” one of the included databases, includes both health & non health care publications

elaboration of how the 3 themes was revealed/synthesized before presenting in the Discussion section

  • We added a paragraph at the end of results section to describe how the three themes were revealed and synthesized. (P 6, line 250)

group the LONG list of activities - starting on line 147, p.4 ff.

  • We grouped the activities per reviewer’s suggestion. (P4-5)

discuss the 3 themes -and probably include considerations related to the motivation of the review stated as "capacity to bring out invisible/tacit", and how considering the 3 themes can give bring in important insights to the health information tech development

  • We added the following sentences to address the reviewer’s concerns: “These themes can provide important insights on HIT development by providing a guidance to researchers and designers to operate and disseminate design workshops in a systematic and effective way.” (Page 11, line 477) and “The success of a design workshop can be measured by its ability to bring out invisible or tacit knowledge. This study provides designers and researchers a variety of options to facilitate capture of this hidden knowledge. As the workshops are more systematically evaluated, which activity is more suitable for different purpose and context will be better understood.” (P12, line 518)
  • elaborate how the section on "Recommendations" comes out of the systematic review - right now it reads more like experiences of the authors.

We added the following statement in the discussion section: “These recommendations were the result of critiquing the literature and discussion with the research team. While this is not a comprehensive list of approaches, it provides our recommendations from the 62 studies. Included studies reported on in-person workshops. However, our recommendations potentially hold for conducting workshops remotely using computer mediation due to pandemics (e.g. COVID19) or any situation that makes gathering impossible or impractical. Novel workshop activities specific to or more effective for remote workshops may be needed.” (P 13, line 591)

Reviewer 2 Report

The present review paper aims to identify possible design approaches for Health Information Technologies, performing an extended search in medical libraries/databases from 2006 to 2020. These good practices are very helpful during the COVID-19 period.

Good approach and methodological description in the text.

It could be helpful if authors were able to provide subgroups of workshops according to the medical field in order to produce some targeted hints to specific medical fields that are for example more "technological".

Consequently, a meta-analysis could be based on the field and not on the type of workshop.

Finally, recommendations could be also provided as a step-by-step approach (or flowing chart), including an effective feedback proposal, for any new workshop designer.

Authors could also recommend that giving voice to the Workshop users as the workshop goes on, this period could provide an additional tool for effective design approaches.

Author Response

The present review paper aims to identify possible design approaches for Health Information Technologies, performing an extended search in medical libraries/databases from 2006 to 2020. These good practices are very helpful during the COVID-19 period. Good approach and methodological description in the text.

  • We thank the reviewer for encouraging words.

It could be helpful if authors were able to provide subgroups of workshops according to the medical field in order to produce some targeted hints to specific medical fields that are for example more "technological".

  • We added a new column in Table 1 entitled “Target Disease, Setting, Treatment or Population” to address the reviewer’s comments. We deleted Publication type column because there are only 5 publications from conference. We just mentioned them in the manuscript.

Consequently, a meta-analysis could be based on the field and not on the type of workshop.

  • Because of the wide variety of conditions addressed in the workshops (Table 1), a meta-analysis was not possible.

Finally, recommendations could be also provided as a step-by-step approach (or flowing chart), including an effective feedback proposal, for any new workshop designer.

  • We revised the order of recommendation to reflect a chronological order and provides a step-by-step approach. Providing a feedback proposal may not be possible because various approaches would be appropriate to collect feedback at the end of workshop. However, we added a few more approaches to collect feedback. (P 12-13)

Authors could also recommend that giving voice to the Workshop users as the workshop goes on, this period could provide an additional tool for effective design approaches.

  • We agree with the reviewer. With the revisions, we discussed the importance of evaluating of workshops by the participants in the discussion section.

Round 2

Reviewer 1 Report

Thank you for revising and include important information to the revision. I still have a couple of major concerns, which is 

  • in the manuscript there is a a "disruptive" flow of the text preceding table 1 and table 2 and it is not possible to follow from the text introducing the tables, with overview of the studies included, with elaboration of evaluation and then the three quite operational topics you emphasize in the discussion
  • it is nice that you included target/setting/type of practice or clinical concern (new column 2) to the overview of the 62 included studies, but on the other hand you are somewhat underutilizing this information and really not including this richness into the discussion topics. Does the literature suggest that it make a difference for PD strategies and why do you not include more of this in workshop recommendations ? 

Author Response

Thank you for revising and include important information to the revision. I still have a couple of major concerns, which is 

  • in the manuscript there is a a "disruptive" flow of the text preceding table 1 and table 2 and it is not possible to follow from the text introducing the tables, with overview of the studies included, with elaboration of evaluation and then the three quite operational topics you emphasize in the discussion

We agree and we revised the organization of text in the Results section to make it flow better.

  • it is nice that you included target/setting/type of practice or clinical concern (new column 2) to the overview of the 62 included studies, but on the other hand you are somewhat underutilizing this information and really not including this richness into the discussion topics. Does the literature suggest that it make a difference for PD strategies and why do you not include more of this in workshop recommendations?

We added a new paragraph (p.4; line 127) into the Results section to better utilize the new column. Although general research question, participant characteristics etc. (as described in [p.13, line 465]) may affect the selection of workshop activities, there is no literature on the affect of target/setting/type of practice or clinical concern on workshop activities. Moreover, the current literature is not mature enough to examine exact selection of workshop activities for specific research question, participant characteristics etc. As more workshops report rigorous evaluations, we will better understand the selection of workshop activities for different disease conditions and other factors.